# Characterization of Molecular Cluster Detection and Evaluation of Cluster Investigation Criteria Using Machine Learning Methods and Statewide Surveillance Data in Washington State

**DOI:** 10.3390/v12020142

**Published:** 2020-01-26

**Authors:** Steven J. Erly, Joshua T. Herbeck, Roxanne P. Kerani, Jennifer R. Reuer

**Affiliations:** 1Office of Infectious Disease, Washington State Department of Health, Olympia, WA 98195, USA; Jennifer.reuer@doh.wa.gov; 2Department of Epidemiology, University of Washington, Seattle, WA 98195, USA; rkerani@uw.edu; 3Department of Global Health, University of Washington, Seattle, WA 98195, USA; herbeck@uw.edu; 4Department of Allergy and Infectious Disease, University of Washington, Seattle, WA 98195, USA

**Keywords:** human immunodeficiency virus (HIV), molecular epidemiology, cluster detection, sequence analysis, public health response, disease surveillance

## Abstract

Molecular cluster detection can be used to interrupt HIV transmission but is dependent on identifying clusters where transmission is likely. We characterized molecular cluster detection in Washington State, evaluated the current cluster investigation criteria, and developed a criterion using machine learning. The population living with HIV (PLWH) in Washington State, those with an analyzable genotype sequences, and those in clusters were described across demographic characteristics from 2015 to2018. The relationship between 3- and 12-month cluster growth and demographic, clinical, and temporal predictors were described, and a random forest model was fit using data from 2016 to 2017. The ability of this model to identify clusters with future transmission was compared to Centers for Disease Control and Prevention (CDC) and the Washington state criteria in 2018. The population with a genotype was similar to all PLWH, but people in a cluster were disproportionately white, male, and men who have sex with men. The clusters selected for investigation by the random forest model grew on average 2.3 cases (95% CI 1.1–1.4) in 3 months, which was not significantly larger than the CDC criteria (2.0 cases, 95% CI 0.5–3.4). Disparities in the cases analyzed suggest that molecular cluster detection may not benefit all populations. Jurisdictions should use auxiliary data sources for prediction or continue using established investigation criteria.

## 1. Introduction

In the past five years, progress in reducing HIV incidence has slowed in the United States [1,2]. In response, the United States federal government launched the Ending the HIV Epidemic Initiative, which sets the goal of reducing new HIV infections by 90% within 10 years. One of the four major components of the initiative is “Respond quickly to potential HIV outbreaks to get needed prevention and treatment services to people who need them” in an effort to reduce transmission [3]. Molecular cluster detection is a component of this.

Molecular epidemiological approaches are used to identify groups of individuals (clusters) with closely related HIV viruses, which may reflect groups of people with ongoing HIV transmission, and lead to prevention resources being directed to those groups [4,5]. The existence of a cluster does not mean that transmission is occurring, however, as some clusters may represent transmission that occurred many years in the past [6]. Similarly, transmission may occur independently of identifiable clustering; incomplete genetic data or late sampling of genetic data (potentially after changes to genetic sequences from selective pressure or genetic drift) can hide genetic linkages [7]. 

While molecular cluster detection can be a useful epidemiological tool, its potential for real-time effectiveness relies partly on the ability to distinguish clusters where transmission is occurring from clusters that are stable. Many state and local health jurisdictions do not have the resources to review and investigate every cluster, therefore prioritization of clusters should be based on the resources available and potential cases averted. The general prioritization criteria suggested by the Centers for Disease Control and Prevention (CDC) for medium and high-burden jurisdictions (>4000 people living with HIV) is to investigate clusters that have grown by five cases linked by a genetic distance of 0.5% (0.005 substitutions per site) in the past 12 months [8]. This criterion is not sensitive to differences in local HIV transmission dynamics, and its appropriateness across jurisdictions or varying resources and priorities is unclear. The Washington State Department of Health currently manually reviews all clusters that have grown by three cases linked by a genetic distance of 1.5% (0.015 substitutions per site) in the last 12 months. Two other investigation criteria are under consideration for use by Washington State: 3 cases in the past 12 months linked by a genetic distance of 0.5% and 5 cases in the past 12 months linked by a genetic distance of 1.5% [9,10].

An ideal cluster investigation criterion would be a strong predictor of cluster growth and would be flexible enough to accommodate varying levels of health department resources. It would also include only data that are readily accessible to surveillance programs to minimize the burden of prediction. Previous efforts to predict cluster growth have been successful in a number of jurisdictions, though some have demonstrated that predictors of cluster growth change in performance over time, which indicates that static models may not be suitable tools for describing a dynamic transmission environment [11,12]. Random forest modeling is a supervised learning algorithm that requires minimal input and has demonstrated considerable predictive success in other public health settings. Unlike traditional regression models, random forests perform well without input such as individual variable selection and fitting [13,14]. The model’s flexibility and ease of implementation make it an appealing choice for the development of a tool that can be tailored to local settings and updated continuously as the context of the HIV epidemic changes. 

The objectives of this study were to characterize current molecular cluster detection and growth in Washington State, to examine the performance of the CDC and current Washington State criteria for cluster review, and to assess the performance of random forest modeling using core surveillance data as a tool to prioritize cluster investigation in Washington State. By identifying successful criteria for cluster investigation, the results of our study can be used to optimize the cluster review and intervention to fully leverage the power of molecular cluster detection to prevent HIV transmission.

## 2. Materials and Methods 

### 2.1. Study Design and Data Collection

We conducted a retrospective study of active clusters of molecularly linked HIV cases in Washington State from 2015 to 2018. Demographic, risk, HIV genetic sequence, and HIV viral load information were collected as part of core HIV surveillance by the Washington State Department of Health. 

Washington State is considered a “moderate prevalence” jurisdiction; in 2018 the HIV prevalence was 13,417 and there were 511 incident cases reported within the state [2,15]. Of these incident cases, 20% had reported previous positive tests and may have been diagnosed in another state and were not in care or were from another country [16]. Washington State has had name-based HIV case reporting and mandatory reporting of CD4 test and all HIV viral load (detectable and undetectable) results since 2006 [15]. HIV genotype sequence reporting was not mandatory during the study period, potentially impacting completeness of molecular cluster membership [17]. Additionally, resistance testing was not ordered for all new HIV diagnoses, also impacting completeness [18]. 

HIV genetic sequences were used to identify clusters of molecularly linked HIV cases using the CDC recommended HIV-TRACE software. Full documentation of the methodology to generate these clusters has been previously described [8,19]. Briefly, the pairwise genetic distances between protease and reverse transcriptase sequences from every pair of individuals was calculated using the Tamura–Nei 93 nucleotide substitution model [20]. For this analysis, we used only the HIV sequence obtained from each individual’s first sample (when HIV sequences were collected from multiple samples of the same individual). Cases were considered molecularly linked if the genetic distance between their sequences was 1.5% or less. Active clusters were defined as HIV clusters of three or more individuals with molecular links in Washington State at any point between 2016 and 2018. Clusters with less than three individuals are not routinely monitored in Washington State.

### 2.2. Variables

On a monthly basis, we described active clusters across the following dimensions: size (number of cases living in Washington), viremic cases (number of cases with last viral load >200 copies/mL or no viral load in last 12 months), late diagnoses (number of cases with first post-diagnosis CD4 count <200 cells/mm^3^), gender (number of female vs. non-female cases), transmission risk (number of cases with reported injection drug use vs. no reported injection drug use), race (number of white cases vs. non-white cases), and time since diagnosis. Cases were determined to be living in Washington from addresses reported to the health department; departure from Washington State was estimated to be at the midpoint between a person’s last reported residence in Washington State and their first reported residence outside of Washington State. The number of viremic cases was assessed using the cases’ most recent viral load value at the current month of analysis. These variables were selected for their availability in surveillance data, associations with transmission rate in Washington State, and their ability to describe historic clusters with rapid growth in Washington State. The variable categories were selected to accommodate population size and to preserve contrasts of particular interest (e.g., transmission risks of male sex with male (MSM), heterosexual contact, pediatric exposure, etc. were all categorized as no injection drug use).

These metrics were expressed as counts, percent of overall cluster size, and percentiles (quartiles for cluster size and viremia, and above and below the median for the number of late diagnosis, gender, transmission risk, and race). To simulate the information available at the time of analysis, cases were included in the cluster based on the date their genotype sequences were received by the Washington State Department of Health.

In addition to point-of-time information, CDC and Washington State candidate criteria for cluster investigation were generated for each month: 3 cases in the previous 12 months at 0.5% genetic distance, 5 cases in the previous 12 months at 0.5% genetic distance, 3 cases in the previous 12 months at 1.5% genetic distance, and 5 cases in the previous 12 months at 1.5% genetic distance (Table 1).

### 2.3. Analysis

To characterize molecular cluster detection in Washington State, we described the state’s molecular cluster detection program in terms of number of active clusters, the proportion of cases (prevalent and incident) with a valid genotype sequence reported to the state, the proportion of cases in any cluster (including historical clusters not analyzed in this study), the proportion of cases in active clusters, the median days from HIV diagnosis to specimen collection of genotype sequences, and the median time from HIV diagnosis to HIV-TRACE analysis.

To characterize the study population, we compared the population of people living with HIV (PLWH) with genotypes (who could potentially be in a cluster) and the population of PLWH in active clusters to the prevalent population of PLWH in Washington State in terms of gender, transmission risk, and age on 31 December 2018.

We measured cluster growth as the number of newly diagnosed cases added to the cluster in the subsequent 3 and 12 months in Washington State, regardless of when the sequence was identified as a member of the cluster. In this case, diagnosis is used as a proxy for transmission, and such cases would be considered potentially preventable during a cluster intervention. Clusters were stratified by posited predictors of cluster growth (cluster size, virally suppressed individuals, % not suppressed, the CDC/Washington State candidate historical criteria, number of cases with a first CD4 count <200, gender, transmission risk, race, and time since diagnosis), which were updated monthly. The number of newly diagnosed cases added to each cluster in the subsequent 3- and 12-month periods was calculated both on an absolute scale and as a rate per 100 person-months. Differences between categories, confidence intervals, and *p*-values were calculated using a repeated-measures generalized estimating equation with a Poisson distribution.

### 2.4. Prediction of Cluster Growth

A random forest regression model was generated to predict the number of newly diagnosed cases added to the cluster in the subsequent three months using data from 2016 to 2017. The following variables were included in the model: number of cases per cluster living in Washington State; number of viremic individuals per cluster; number of new cases in the previous 1, 3, and 12 months at both 0.015 and 0.005 genetic distance; number of late diagnoses; number of white cases; number of cases described as injection drug use transmission risk; number of female cases; and number of cases that had been diagnosed <1 year prior. A random search was used to identify optimum number of candidate variables at each split. Five hundred trees were estimated. A second model was fit to predict rate of cluster growth per 100 person-months. Regression was performed using the RandomTree package in R [21].

To create an investigation criterion from this model, we derived a cutoff for predicted cluster growth at which an investigation should be initiated. For comparability with the existing criteria, we sought a criterion that would yield approximately the same number of investigations as the existing criteria. To identify this target number of investigations, we counted the number of investigations initiated by the existing criteria from 2016 to 2017 (*n*). To translate this into a cutoff for the random forest models, we ranked the clusters by their highest monthly predicted growth from the random forest model. The predicted growth of the top *n*th cluster was selected as the cutoff for investigation, as this value would yield *n* investigations in 2016–2017.

Using this technique, we derived two investigation criteria to match the activity of the most stringent and least stringent CDC and Washington State investigation criteria (five cases at 0.005% genetic distance and three cases at 0.015% genetic distance). 

### 2.5. Evaluation of Investigation Criteria

The CDC cluster investigation criterion, Washington State cluster investigation criterion, and random forest criteria were evaluated for 2018 on the basis of growth of clusters indicated for investigation. The number of new diagnoses in the subsequent three months and the rate of new diagnoses per 100 person-months were calculated for clusters that met each criterion. To simulate a real public health intervention, clusters were only eligible for investigation once. Criteria that identified clusters with higher growth over the subsequent three-month period were considered the best.

The data collected and analyses conducted under HIV/AIDS surveillance authority as an evaluation of the surveillance system are not considered research. No data which could identify individuals are presented.

## 3. Results

### 3.1. Cluster Detection in Washington State

Between 2015 and 2018, 57% (1058 of 1847) of new cases had a reverse transcriptase or protease sequence reported to the state. Only 49% (7373 of 15,150) of prevalent cases in that time frame had sequences. The median days from diagnosis to genotype specimen collection was 14 (IQR 6–31) and from diagnosis to TRACE analysis was 291 (138–714). From 2015 to 2018, these delays decreased from a median of 17 to 11 days and 821 to 109 days, respectively (Table 2). People with a genotype were more likely to be under 45 years of age (41% vs. 33%, *p* < 0.01) (Table 3).

Using the 1.5% genetic distance threshold, 107 clusters were present during the study time period, which represented 22% of new HIV cases and 8% of prevalent HIV cases (Table 2). Compared to the total population of PLWH in Washington, those within clusters of three or more prevalent cases were more likely to be recently diagnosed (47% diagnosed in the last five years vs. 21%), male (93% vs. 83%, *p* < 0.01), white (64% vs. 57%, *p* < 0.01), be described as MSM transmission risk (74% vs. 60%, *p* < 0.01), and be under 45 years old (68% vs. 34%, *p* < 0.01) (Table 3).

### 3.2. Predictors of Cluster Growth

In 2016 and 2017, the mean number of new cases added to clusters was 0.24 (95% CI 0.18–0.33) per three-month period and 1.22 (95% CI 0.86–1.73) per 100 person-months. Over a three-month period, larger clusters and clusters with more viremic individuals grew faster on an absolute scale. Clusters with three or fewer members grew at an average of 0.17 cases (95% CI 0.11–0.17), while clusters with 12 or more members grew at an average of 0.37 cases (95% CI 0.24–0.58). Clusters with zero viremic individuals grew at an average of 0.13 cases (95% CI 0.08–0.20), while clusters with three or more viremic individuals grew at an average of 0.38 cases (95% CI 0.26–0.57). As a rate, the only significant predictor of growth was cluster size, where larger clusters grew more slowly (0.43, 95% CI 0.33–0.58 cases per 100 person-months in clusters of 12 members or more vs. 2.07, 95% CI 1.29–3.34 cases in clusters with three or fewer members).

Clusters meeting CDC criteria of five cases in the previous 12 months at a genetic distance of 0.005 had the highest growth on the absolute scale (1.50, 95% CI 0.79–2.86) and a high growth rate (2.93 per 100 person-months, 95% CI 0.79–10.86), but these values were not significantly different from clusters outside of this criteria (Table 4). Assessed at 12 months of growth, clusters showed similar, but attenuated trends (Table 5).

### 3.3. Derivation and Evaluation of Investigation Criteria

The CDC criteria identified four clusters from 2016 to 2017 that would prompt investigation. To create a criterion that would produce a similar volume of investigations, a cutoff was selected for the random forest model (i.e., the predicted number of newly diagnosed cases in the subsequent three months that would prompt investigation). The criterion to prompt investigation was set at 2.3 predicted cases. This value would have initiated the same number of investigations on unique clusters from 2016 to 2017. A looser criterion was set at 0.9 predicted cases, as this would have matched the 24 clusters investigated under the Washington State Department of Health investigation criterion. For growth per 100 person-months, criteria of 0.14 and 0.42 were selected. The first four layers of an example tree sampled from the random forest are presented in Figure 1. Of note, this represents one of the 500 trees which were combined to create the prediction model.

In 2018, there were six clusters that met the CDC investigation criterion, which had on average 28.3 (95% CI 5.7–62.4) members and grew 2.0 (95% CI 0.5–3.4) members in the subsequent three months. The clusters selected for investigation by the random forest model did not grow significantly faster. The model selected four clusters with a mean of 31.8 (95% CI 20.5–42.9) members that grew 2.3 (95% CI 1.3–3.2) members in the subsequent three months. There were 17 clusters that met the Department of Helath investigation criterion that had a mean size of 20.5 (95% CI 8.3–32.6) and grew an average of 1.4 cases (95% CI 0.8–2.0). The clusters selected by the random forest model with a looser criterion did not grow significantly faster (*n* = 15, mean size = 19.9, 95% CI 16.8–23.1, mean growth = 1.3, 95% CI 1.1–1.4). All criteria selected clusters that grew significantly faster than the statewide cluster growth of 0.3 cases per three months.

Prediction of cluster growth per 100 person-months was poor, and only the random forest with the looser criterion selected clusters that grew significantly faster than the statewide cluster growth of 1.4 cases per 100 person-months. This model detected 20 clusters with a mean size of 7.5 (95% CI 6.1–8.8) and a mean growth of 3.4 (95% CI 2.8–4.0) per 100 person-months. The random forest model with the stricter criterion only identified one cluster, which did not grow (Table 6).

## 4. Discussion

Molecular cluster detection of HIV has the potential to guide public health intervention to populations of the greatest need. This is dependent on completion of genetic sequence data and the ability to identify molecular clusters where transmission is likely to occur. In this analysis, we demonstrated the disparities in the applicability of molecular cluster detection in populations in Washington State and the challenge of predicting cluster growth from core surveillance data. Finally, we suggest that the established criteria for prediction of cluster growth based on cases in the previous 12 months continue to be used in absence of additional data sources.

Molecular cluster detection is dependent on the timeliness and completeness of the sequence data included in the analysis [17]. From 2015 to 2018, there was an 87% decrease in the median time from a person’s HIV diagnosis to analysis in HIV-TRACE. In the same time period, Washington State was near the CDC target of 60% completeness of sequence data in new HIV cases (57%) but only achieved 49% in prevalent cases. The majority of incident and prevalent cases do not have an observed molecular link within the state (71% and 82%, respectively.) While the population of PLWH with analyzable genotype sequences was fairly equivalent to the population of Washington State, there were stark disparities in the populations who were in analyzable clusters of three or more individuals. People in analyzable clusters are more likely to be white, male, and MSM. In Washington State, these are populations with decreasing incidence counts and a higher proportion of viral suppression, which may indicate that these populations may not be those who could most benefit from interventions informed by molecular cluster detection. The disparity between the population with analyzable genotype sequences and the population in analyzable clusters may be tied to disparity in time since diagnosis, which could hide a difference in the distribution of cases with sequences in more recent years. There may also be differences in the network structure of demographic groups that affect the number of analyzable clusters.

Completeness of collection of molecular sequences can be a challenge for health departments as there are multiple points in the process that can result in a sequence not being available for analysis: an individual is not diagnosed, the individual is not linked to care, a resistance test is not ordered at the time of initial linkage to care, there is a problem with the specimen collection, there is a problem with specimen storage or transportation, there is a problem at the time of running the test, or the sequence is not transmitted to the health department. Efforts to improve the completeness (to increase the percentage of new diagnoses with reported sequences) and efficiency (to decrease the amount of time for inclusion of sequences in analysis) of the process are important to the utility of the program.

The random forest model used in this analysis did not predict cluster growth significantly better than the CDC or Washington State criteria. This demonstrates the limitations in predictive power of HIV core surveillance and molecular cluster detection in isolation. Previous successful prediction efforts have used partner services data to supplement these information sources, but some jurisdictions may have limited access to such data through structural or legal barriers [12]. In addition to partner services data, information about cluster topology and data from prior cluster and data for care investigations could be useful in prediction of cluster growth. The performance of our prediction model (relative to the existing criteria) was likely hurt by the rapid growth of an outbreak cluster among heterosexual persons who were living homeless and injecting drugs in King County in 2018, which was unlike the baseline cluster growth in years past [10]. This is the type of event that would be important to predict, however, and highlights the limitations of prediction models in the domain of rare events [22]. A larger dataset, either covering a longer time period or a larger geographic region would be useful for predicting such events.

In the absence of complete or readily available adjunct data from partner services investigations, the results of this study suggest that jurisdictions use the established criteria based on cluster growth in the past 12 months. Looser criteria (e.g., smaller numbers of individuals at a less restrictive genetic distance cutoff) may offer an added benefit of offering staff a chance to identify non-surveillance indicators that are important in the local transmission networks.

Notably, no criteria examined in our analyses was able to predict cluster growth as a rate. Jurisdictions that are concerned with cases prevented as a function of investigated cases may need to seek criteria based on information outside of core surveillance.

For cluster detection and response to be most effective it should incorporate robust core surveillance including an ongoing understanding of changes in HIV transmission trends and complete case and lab reporting, robust partner services activities with disease investigators that are familiar with the profile of their local cases, and efficient and complete molecular data collection and analysis [10,12,23]. There should also be transparency and dialogue with the community about the activities being undertaken and why they are important. It is only with the combination of information and efforts that programs will be able to make effective use of cluster detection to decrease HIV transmission early and focus prevention efforts.

## Figures and Tables

**Figure 1 viruses-12-00142-f001:**
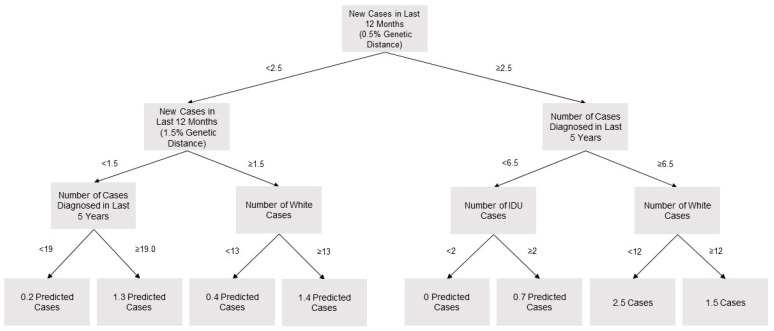
Four layer sample tree from 500-tree random forest model to predict three-month cluster growth, Washington State 2016–2017.

**Table 1 viruses-12-00142-t001:** Centers for Disease Control and Prevention (CDC) and proposed Washington State criteria for cluster investigation.

Investigation Criteria	Genetic Distance Threshold	Number of Diagnoses in Past 12 Months
Washington State (Loose)	1.5%	3
Washington State (Strict)	1.5%	5
CDC (Strict)	0.5%	5

**Table 2 viruses-12-00142-t002:** Summary of molecular cluster detection completion and timeliness in Washington State, 2015–2019.

Variable	2015	2016	2017	2018	Total
Number of Analyzed Clusters ^a^	95	95	103	107	107
Number of New Cases WA State	459	436	441	511	1847
New Cases with Analyzable Genotype ^b^	276 (60%)	238 (55%)	252 (57%)	292 (57%)	1058 (57%)
Number of New Cases in All Clusters	156 (34%)	112 (26%)	134 (30%)	148 (29%)	550 (29%)
Number of New Cases in Analyzed Clusters	85 (19%)	84 (19%)	112 (25%)	120 (23%)	401 (22%)
Cumulative Prevalent Cases in WA State ^c^	12,631	13,051	13,424	13,764	15,150
Prevalent Cases with Analyzable Genotype ^b^	6320 (50%)	6458 (49%)	6583 (49%)	6733 (49%)	7373 (49%)
Number of Prevalent Cases in All Clusters	2238 (18%)	2310 (18%)	2389 (18%)	2476 (18%)	2680 (18%)
Number of Prevalent Cases in Analyzed Clusters	824 (7%)	897 (7%)	983 (7%)	1078 (8%)	1157 (8%)
Median Days from Dx to Specimen Collection	17 (6–30)	14 (7–31)	15 (7–31)	11 (4–33)	14 (6–31)
Median Days from Dx to TRACE Analysis	821 (734–921)	468 (353–576)	201 (143–255)	109 (77–137)	291 (138–714)

^a^ Analyzed clusters included members of molecularly linked clusters (genetic distance 0.015) with three or more prevalent cases at one point in time between 2016 and 2018. ^b^ Analyzable genotypes include reverse transcriptase and protease sequences. ^c^ Cumulative prevalent cases include all people with diagnosed HIV who lived in Washington State at some point during the time period.

**Table 3 viruses-12-00142-t003:** Demographic characteristics of study population, prevalent cases, and cases with genotypes WA State 2015–2018 ^a^.

Variable	Value	All Cases	Cases with PR or RT Sequence	*p*-Value ^b^	Study Population ^a^	*p*-Value ^b^
N		15,150	7499	-	1157	-
Time Since Diagnosis	<1 Year	540 (4%)	297 (4%)	<0.01	121 (11%)	<0.01
	1–5 Years	2557 (17%)	1393 (19%)		417 (36%)	
	6–10 Years	2870 (19%)	1797 (24%)		433 (37%)	
	>10 Years	7846 (51%)	3886 (53%)		186 (16%)	
Gender	Female	2278 (15%)	1108 (15%)	<0.01	73 (6%)	<0.01
	Transgender Male	14 (0%)	7 (0%)		0 (0%)	
	Male	12,727 (84%)	6173 (83%)		1072 (93%)	
	Transgender Female	131 (1%)	85 (1%)		12 (1%)	
Race	WHITE	8747 (58%)	4046 (55%)	<0.01	742 (64%)	<0.01
	BLACK	2588 (17%)	1273 (17%)		99 (9%)	
	HISP	2152 (14%)	1118 (15%)		194 (17%)	
	ASIAN	516 (3%)	248 (3%)		31 (3%)	
	HAW/PI	69 (0%)	43 (1%)		10 (1%)	
	AI/AN	156 (1%)	81 (1%)		12 (1%)	
	MULTI	915 (6%)	564 (8%)		69 (6%)	
	UNK	7 (0%)	0 (0%)		0 (0%)	
Age (31 December, 2018)	<13	40 (0%)	12 (0%)	<0.01	0 (0%)	<0.01
	13–24	328 (2%)	171 (2%)		56 (5%)	
	25–34	2143 (14%)	1143 (16%)		385 (33%)	
	35–44	3089 (20%)	1680 (23%)		340 (29%)	
	45–54	4456 (29%)	2219 (30%)		250 (22%)	
	55–64	3728 (25%)	1677 (23%)		108 (9%)	
	>64	1366 (9%)	471 (6%)		18 (2%)	
Risk	MSM	9234 (61%)	4392 (60%)	<0.01	854 (74%)	<0.01
	IDU	924 (6%)	502 (7%)		79 (7%)	
	MSM/IDU	1465 (10%)	813 (11%)		137 (12%)	
	TRANFUS	21 (0%)	9 (0%)		0 (0%)	
	HEMO	21 (0%)	6 (0%)		0 (0%)	
	HETERO	1787 (12%)	875 (12%)		38 (3%)	
	PED	140 (1%)	73 (1%)		0 (0%)	
	NIR	1549 (11%)	703 (10%)		(4%)	

PR = Protease, RT = Reverse Transcriptase. ^a^ Study population included members of molecularly linked clusters (genetic distance 0.015) with three or more prevalent cases at one point in time between 2015 and 2018. ^b^
*p*-values from chi-square test comparing listed population to the remainder of all cases.

**Table 4 viruses-12-00142-t004:** Cluster growth rates in subsequent three-month period by cluster attribute, Washington State 2016–2017.

Variable	Value	Cluster-Months	Absolute 3 Month Cluster Growth, Mean (5% CI) ^a^	*p*-Value	Cluster Growth Per 100 Person-Months, Mean (95% CI) ^a^	*p*-Value
Total Population	All	2318	0.24 (0.18–0.33)		1.22 (0.86–1.73)	
Viremic Individuals	0	344	0.13 (0.08–0.20)	0.023	1.20 (0.72–2.01)	0.212
	1	533	0.22 (0.15–0.33)		1.69 (1.04–2.76)	
	2	462	0.18 (0.12–0.27)		1.00 (0.65–1.55)	
	3+	530	0.38 (0.26–0.57)		0.96 (0.55–1.68)	
Percent Viremic (Quartiles)	<12%	353	0.12 (0.08–0.20)	0.024	1.17(0.70–1.96)	0.248
	12–25%	692	0.32 (0.23–0.46)		0.92 (0.63–1.35)	
	26–35%	496	0.23 (0.15–0.35)		1.19 (0.76–1.88)	
	≥35%	328	0.21 (0.13–0.36)		1.97 (1.05–3.69)	
Cluster Size (Quartiles)	≤3	517	0.17 (0.11–0.27)	0.010	2.07 (1.29–3.34)	0.011
	4–5	430	0.13 (0.08–0.20)		0.93 (0.60–1.45)	
	6–12	498	0.30 (0.19–0.47)		1.26 (0.80–2.00)	
	>12	424	0.37 (0.24–0.58)		0.43 (0.33–0.58)	
% White	<16%	939	0.23 (0.15–0.35)	0.685	1.18 (0.71–1.95)	0.774
	≥18%	930	0.25 (0.18–0.35)		1.27 (0.88–1.85)	
% Female	0%	1343	0.21 (0.15–0.31)	0.203	1.18 (0.76–1.82)	0.727
	>0%	526	0.31 (0.21–0.47)		1.34 (0.76–2.35)	
% IDU	<16%	943	0.24 (0.17–0.35)	0.966	0.93 (0.70–1.23)	0.165
	≥15%	926	0.24 (0.15–0.38)		1.52 (0.90–2.58)	
% Late	<13%	932	0.24 (0.16–0.35)	0.933	1.08 (0.73–1.62)	0.522
	≥13%	937	0.24 (0.16–0.38)		1.36 (0.80–2.33)	
% Diagnosed in Past 5 Years	<25%	363	0.19 (0.13–0.29)	0.721	1.05 (0.56–1.98)	0.572
	25–50%	702	0.26 (0.17–0.40)		1.09 (0.67–1.79)	
	51–66%	349	0.26 (0.17–0.38)		1.45 (0.95–2.21)	
	>66%	455	0.24 (0.13–0.46)		1.39 (0.85–2.25)	
3 Cases in Previous 12 Months (0.015)	No	1685	0.18 (0.15–0.22)	0.022	1.09 (0.83–1.41)	0.253
	Yes	184	0.78 (0.47–1.27)		2.49 (0.98–6.33)	
5 Cases in Previous 12 Months (0.015)	No	1819	0.21 (0.17–0.27)	0.088	1.16 (0.86–1.56)	0.263
	Yes	50	1.26 (0.89–1.78)		3.69 (1.49–9.14)	
3 Cases in Previous 12 Months (0.005)	No	1818	0.21 (0.17–0.27)	0.058	1.17 (0.85–1.62)	0.224
	Yes	51	1.29 (0.74–2.26)		3.00 (1.19–7.58)	
5 Cases in Previous 12 Months (0.005)	No	1847	0.23 (0.17–0.29)	0.116	1.20 (0.86–1.68)	0.388
	Yes	22	1.50 (0.79–2.86)		2.93 (0.79–10.86)	

IDU = Injection Drug Use Transmission Risk. ^a^ Calculated from number of newly diagnosed cases joining the cluster in the subsequent three-month period. Estimates, 95% confidence intervals, and *p*-values from repeated-measures generalized estimating equation using a Poisson distribution.

**Table 5 viruses-12-00142-t005:** Cluster growth rates in subsequent 12-month period by cluster attribute, Washington State 2016–2017.

Variable	Value	Cluster-Months	Absolute 12-Month Cluster Growth, Mean (95% CI) ^a^	*p*-value	Cluster Growth Per 100 Person-Months, Mean (95% CI) ^a^	*p*-Value
Total Population	All	2318	1.02 (0.75–1.38)		1.27 (0.89–1.81)	
Viremic Individuals	0	344	0.62 (0.31–1.25)	0.081	1.35 (0.60–3.05)	0.369
	1	533	0.85 (0.59–1.23)		1.61 (1.07–2.42)	
	2	462	0.82 (0.57–1.20)		1.18 (0.78–1.81)	
	3+	530	1.62 (1.09–2.41)		0.95 (0.56–1.62)	
Percent Viremic (Quartiles)	<12%	353	0.65 (0.34–1.25)	0.216	1.33 (0.59–2.98)	0.056
	12%–25%	692	1.28 (0.87–1.87)		0.83 (0.56–1.22)	
	26%–35%	496	0.95 (0.65–1.39)		1.40 (0.95–2.07)	
	≥35%	328	0.98 (0.55–1.77)		1.94 (1.06–3.53)	
Cluster Size (Quartiles)	≤3	517	0.74 (0.48–1.17)	0.029	2.14 (1.37–3.34)	0.014
	4–5	430	0.53 (0.33–0.84)		0.94 (0.60–1.49)	
	6–12	498	1.28 (0.79–2.09)		1.37 (0.81–2.32)	
	>12	424	1.55 (0.96–2.50)		0.42 (0.32–0.55)	
% White	<16%	939	0.87 (0.58–1.20)	0.237	1.15 (0.71–1.84]	0.521
	≥18%	930	1.18 (0.80–1.72)		1.40 (0.89–2.20)	
% Female	0%	1343	0.83 (0.57–1.22)	0.072	1.04 (0.69–1.55)	0.184
	>0%	526	1.50 (0.98–2.30)		1.87 (1.02–3.41)	
% IDU	<16%	943	0.99 (0.66–1.48)	0.836	0.84 (0.63–1.11)	0.050
	≥15%	926	1.05 (0.68–1.62)		1.71 (1.05–2.78)	
% Late	<13%	932	1.03 (0.68–1.57)	0.940	1.11 (0.66–1.86)	0.472
	≥13%	937	1.01 (0.66–1.55)		1.43 (0.89–2.31)	
% Diagnosed in Past 5 Years	<25%	363	0.87 (0.54–1.39)	0.670	1.28 (0.55–2.97)	0.143
	25–50%	702	1.02 (0.63–1.65)		0.94 (0.61–1.45)	
	51–66%	349	1.27 (0.83–1.96)		1.96 (1.22–3.15)	
	>66%	455	0.95 (0.49–1.87)		1.24 (0.70–2.19)	
3 Cases in Previous 12 Months (0.015)	No	1685	0.82 (0.64–1.05)	0.029	1.15 (0.85–1.58)	0.252
	Yes	184	2.88 (1.77–4.69)		2.33 (1.01–5.42)	
5 Cases in Previous 12 Months (0.015)	No	1819	0.91 (0.7–1.19)	0.105	1.20 (0.86–1.67)	0.337
	Yes	50	4.98 [3.28–7.55)		3.75 (1.32–10.67)	
3 Cases in Previous 12 Months (0.005)	No	1818	0.91 (0.7–1.18)	0.041	1.20 (0.86–1.68)	0.154
	Yes	51	5.12 [3.4–7.69)		3.74 (1.75–8.00)	
5 Cases in Previous 12 Months (0.005)	No	1847	0.97 (0.73–1.29)	0.098	1.26 (0.89–1.79)	0.507
	Yes	22	4.91 (2.7–8.92)		2.11 (0.62–7.19)	

IDU = Injection Drug Use Transmission Risk.^a^ Calculated from number of newly diagnosed cases joining the cluster in the subsequent 12-month period. Estimates, 95% confidence intervals, and *p*-values from repeated-measures generalized estimating equation using a Poisson distribution.

**Table 6 viruses-12-00142-t006:** Comparison of investigation criteria by cluster size and growth at first indicated investigation, Washington State 2018.

**Cluster Investigation Criteria**	**Number of Clusters Reaching Investigation Criteria**	**Prevalent Cases at First Indicated Investigation, Mean (95% CI)**	**Absolute 3 Month Cluster Growth at First Indicated Investigation, Mean (95% CI) ^a^**
3 Cases at 0.015	17	20.5 (8.3–32.6)	1.4 (0.8–2.0)
5 Cases at 0.015	9	28.0 (4.9–50.2)	1.4 (0.5–2.3)
3 Cases at 0.005	10	24.6 (6.6–42.6)	1.6 (0.8–2.4)
5 Cases at 0.005	6	28.3 (−5.7–62.4)	2.0 (0.5–3.4)
>0.9 Predicted Cases in 3 Months (Random Forest) ^b^	14	19.9 (16.8–23.1)	1.3 (1.1–1.4)
>2.3 Predicted Cases in 3 Months (Random Forest) ^b^	4	31.8 (20.5–42.9)	2.3 (1.3–3.2)
**Cluster Investigation Criteria**	**Number of Clusters Reaching Investigation Criteria**	**Prevalent Cases at First Indicated Investigation, Mean (95% CI)**	**Cluster Growth Per 100 Person-Months at First Indicated Investigation, Mean (95% CI) ^a^**
3 Cases at 0.015	17	20.5 (8.3–32.6)	1.5 (1.3–7.7)
5 Cases at 0.015	9	28.0 (4.9–50.2)	2.6 (0.7–4.4)
3 Cases at 0.005	10	24.6 (3.8–45.4)	5.3 (−0.3–10.9)
5 Cases at 0.005	6	28.3 (−5.7–62.4)	3.1 (0.8–5.4)
>0.14 Predicted Cases Per 100 Person-Months (Random Forest) ^b^	15	7.5 (6.1–8.8)	3.4 (2.8–4.0)
>0.42 Predicted Cases Per 100 Person-Months (Random Forest) ^b^	1	2 (NA)	0 (NA)

^a^ Calculated from number of newly diagnosed cases joining the cluster in the subsequent 12-month period, 95% confidence intervals from normal distribution. ^b^ Random forest model with 500 trees and 6 variables selected at each branch fit to 2016 and 2017 cluster data. The models included the following variables: prevalent cases; number of cases diagnosed in the previous year; number of viremic cases; number of cases in past 1, 3, and 12 months (0.015 and 0.005 genetic distance); number of late diagnoses; number of white cases; number of female cases; and number of cases with injection drug transmission risk. Cutoffs were selected approximate the same number of investigations as the “3 Cases at 0.015” and “5 Cases at 0.005” criteria.

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
