# Peer review of "Characterization of Molecular Cluster Detection and Evaluation of Cluster Investigation Criteria Using Machine Learning Methods and Statewide Surveillance Data in Washington State"

_viruses, 2020, doi:10.3390/v12020142_

Round 1
Reviewer 1 Report
The article is very well written and is very informative.
The methods section is appropriate for the objectives of the study, the results are clearly described, and the discussions are in line with the findings of the study.
I recommend accepting this paper after a few minor revisions:
Page 2 - line 91 - 92: "Additionally, resistance testing is not ordered ... impacting completeness [17]". Please put the sentence in past tense - to be consistent with the rest of the text. The meaning of the abbreviation PLWH has been provided the first time it was used in the abstract; however, it was not described the first time the same abbreviation was used in the text. Please, do the same in page 3 - line 132. Page 4 - line 178: "..evaluation of the surveillance system and isnot considered research." - Please, correct the typo. Page 12 - line 272: "HIV core sruveillance and molecular cluster detection in isolation" - please, correct the typo.Author Response
Thank you for your review of our manuscript! I have corrected the two spelling errors, tense issue, and abbreviation definition identified in the review.
Reviewer 2 Report
Summary
This article leverages machine learning methods to evaluate molecular cluster detection and investigation criteria used by Washington State, the CDC, and other US jurisdictions. Through a retrospective evaluation of rapidly growing HIV transmission clusters and investigations thereof, the results independently recapitulate the investigative guidance espoused by the CDC. Importantly, these results are derived from similar data sources thereby identifying a potential need to adopt auxiliary data sources.
General:
The authors reference hidden links (line 45) and that only a fraction of incident/prevalent cases (lines 250-251) link to another sequence. It may benefit the public health and research communities to recapitulate this finding in another way: that the overwhelming majority of incident and prevalent cases (71% and 82%, respectively) do not have an observed molecular link within the state.
The core finding, that jurisdictions should use auxiliary data sources or continue with the status quo should be accompanied by suggested data sources. For example, quantitative characteristics of cluster topology (centrality, clustering coefficients, etc) may provide a fruitful means of differentiating rapid growth clusters. Another example might be data sources related to prior cluster investigations and their outcomes.
Visualization:
For visualization and communication purposes, the authors should consider rendering a tree sampled from the forest as a decision tree classifier for the data set. This would provide a framework from which an uninitiated reader can understand the more robust random forest approach.
Consider a cluster/network visualization where clusters are colored according to whether it was flagged by one or more of the criteria under evaluation.
Comments:
Lines 50-59: The authors outline prioritization criteria for the CDC and Washington State. Given the critical role that these criteria play in the manuscript, the reader might benefit from their instantiation as a table or figure.
Line 66: Consider providing a citation where random forest and decision tree modeling have been leveraged in the realm of public health and/or HIV.
Lines 100-101: It is not clear why active clusters were defined as 3 or more individuals with molecular links in Washington State between 2016 and 2018. Was this defined by Washington State, CDC, or the authors?
Line 103: “month-by month” to “month-by-month” or “monthly”
Lines 103-108: This section lists variables and their definitions. For example, gender is encoded as ‘female vs non-female cases’. Later, the authors state that these metrics were expressed as counts, percentages and percentiles.
Lines 114-115: The authors state that “Variables were collapsed into larger categories due to limitations of populations size and variability.” This sentence is vague, given its position in the paragraph. Consider moving it closer to the collapsed variables that it references. Also consider providing an example of a collapsed variable (e.g., MSM, heterosexual, etc à no reported injection drug use). The reader may also benefit from a more complete description of criteria used to determine the best categories to collapse.
Lines 185-186: The reported finding of a drastic reduction in delay for genotyping and HIV-TRACE analysis is drastically understated. Authors should consider highlighting this finding in a figure or again in the discussion.
Author Response
Thank you for taking the time to carefully go through and review our manuscript. Please see the attachment for a point-by-point response.
General:
Point 1: The authors reference hidden links (line 45) and that only a fraction of incident/prevalent cases (lines 250-251) link to another sequence. It may benefit the public health and research communities to recapitulate this finding in another way: that the overwhelming majority of incident and prevalent cases (71% and 82%, respectively) do not have an observed molecular link within the state.
Thank you for your feedback! We have reversed the phrasing of this sentence as you suggested (Lines 264)
Point 2: The core finding, that jurisdictions should use auxiliary data sources or continue with the status quo should be accompanied by suggested data sources. For example, quantitative characteristics of cluster topology (centrality, clustering coefficients, etc) may provide a fruitful means of differentiating rapid growth clusters. Another example might be data sources related to prior cluster investigations and their outcomes.
We have added the sentence “In addition to partner services data, information about cluster topology and data from prior cluster and data to care investigations could be useful in prediction of cluster growth” to line 288
Visualization:
Point 3: For visualization and communication purposes, the authors should consider rendering a tree sampled from the forest as a decision tree classifier for the data set. This would provide a framework from which an uninitiated reader can understand the more robust random forest approach.
We have added a figure and explanation to the manuscript (Line 230)
Point 4: Consider a cluster/network visualization where clusters are colored according to whether it was flagged by one or more of the criteria under evaluation.
We are under advisement by the CDC to avoid dissemination of genetic network diagrams due to their potential for misinterpretation as transmission diagrams. The use of molecular surveillance by public health agencies is a sensitive topic in some jurisdictions (especially those with HIV criminalization laws), and there is some concern that these types of visualizations promote misconceptions about the ways that genetic data can be used. For more information, see Detecting and Responding to HIV Transmission Clusters: A guide for Health Departments, Page 13
Comments:
Point 5: Lines 50-59: The authors outline prioritization criteria for the CDC and Washington State. Given the critical role that these criteria play in the manuscript, the reader might benefit from their instantiation as a table or figure.
We added a table to highlight these definitions (Line 129)
Point 6: Line 66: Consider providing a citation where random forest and decision tree modeling have been leveraged in the realm of public health and/or HIV.
We have added a reference to Kane et al’s work on predicting avian influenza outbreaks using random forest models:
Kane, M.J., Price, N., Scotch, M. et al. Comparison of ARIMA and Random Forest time series models for prediction of avian influenza H5N1 outbreaks. BMC Bioinformatics 15, 276 (2014) doi:10.1186/1471-2105-15-276
(Line 69)
Point 7: Lines 100-101: It is not clear why active clusters were defined as 3 or more individuals with molecular links in Washington State between 2016 and 2018. Was this defined by Washington State, CDC, or the authors?
We have added a sentence clarifying that clusters with less than three individuals are not routinely monitored in Washington State (Line 101)
Point 8: Line 103: “month-by month” to “month-by-month” or “monthly”
Changed to monthly (Line 104)
Point 9: Lines 103-108: This section lists variables and their definitions. For example, gender is encoded as ‘female vs non-female cases’. Later, the authors state that these metrics were expressed as counts, percentages and percentiles.
The wording of this section was updated for structural consistency (Line 105)
Point 10: Lines 114-115: The authors state that “Variables were collapsed into larger categories due to limitations of populations size and variability.” This sentence is vague, given its position in the paragraph. Consider moving it closer to the collapsed variables that it references. Also consider providing an example of a collapsed variable (e.g., MSM, heterosexual, etc à no reported injection drug use). The reader may also benefit from a more complete description of criteria used to determine the best categories to collapse.
We reworded the sentence to make its context more clear and to provide an example. The section now reads:
These variables were selected for their availability in surveillance data, associations with transmission rate in Washington State, and their ability to describe historic clusters with rapid growth in Washington State. The variable categories were selected to accommodate population size and to preserve contrasts of specific particular interest (e.g, transmission risks of MSM, heterosexual contact, pediatric exposure, etc. were all categorized as no injection drug use). (Line 115)
Point 11: Lines 185-186: The reported finding of a drastic reduction in delay for genotyping and HIV-TRACE analysis is drastically understated. Authors should consider highlighting this finding in a figure or again in the discussion.
We have added a sentence to the discussion highlighting this 87% decrease (Line 261)